# Slicing Spheroids in Microfluidic Devices for Morphological and Immunohistochemical Analysis

**DOI:** 10.3390/mi11050480

**Published:** 2020-05-06

**Authors:** Satoru Kuriu, Tetsuya Kadonosono, Shinae Kizaka-Kondoh, Tadashi Ishida

**Affiliations:** 1Department of Mechanical Engineering, School of Engineering, Tokyo Institute of Technology, Kanagawa 226-8503, Japan; 2Department of Life Science and Technology, School of Life Science and Technology, Tokyo Institute of Technology, Kanagawa 226-8503, Japan; tetsuyak@bio.titech.ac.jp (T.K.); skondoh@bio.titech.ac.jp (S.K.-K.)

**Keywords:** microfluidic device, embedding resin, spheroid, sectioning, immunohistochemistry

## Abstract

Microfluidic devices utilizing spheroids play important roles in in vitro experimental systems to closely simulate morphological and biochemical characteristics of the in vivo tumor microenvironment. For the observation and analysis of the inner structure of spheroids, sectioning is an efficient approach. However, conventional microfluidic devices are difficult for sectioning, and therefore, spheroids inside the microfluidic channels have not been sliced well. We proposed a microfluidic device created from embedding resin for sectioning. Spheroids were cultured, embedded by resin, and sectioned in the microfluidic device. Slices of the sectioned spheroids yielded clear images at the cellular level. According to morphological and immunohistochemical analyses of the slices of the spheroid, specific protein distribution was observed.

## 1. Introduction

Spheroids are scaffold-free spherical cell aggregates that are widely used as models that can reproduce the characteristics of various tumors in vivo [1,2,3]. The cells within a given spheroid form cell-to-cell and cell-to-extra-cellular matrix structures that also exist in vivo, while monolayer cultured cells do not form these structures [1,2]. These structures work as chemical or physical barriers against treatments using anticancer drugs or nanoparticles, respectively [4,5]. By utilizing spheroids, relevant data regarding the in vivo environment can be acquired in vitro. However, conventional experimental methods using culture dishes or well plates cannot apply flow to the spheroids and cannot insert blood vessels into the spheroid, and they are therefore not sufficient to mimic the environment of real tumors. Based on the recent advances in microfluidic techniques, a vascularized spheroid was formed in a microchannel by generating a concentration gradient of vascular endothelial growth factor [6]. Dynamic flow behaviors, including sheer stress that is important for drug delivery efficacy and cellular responses [7], can be applied to the cells within the spheroids through vascularized vessels [6].

Optical microscopy is utilized for the analysis of spheroids. The surface structures and molecular distributions are analyzed using bright-field and fluorescent images, respectively. However, dense cellular structures within the spheroids scatter and absorb incident and fluorescent lights. The inner structures of the spheroids are difficult to visualize [8,9]. To visualize the inner structures, tissue-clearing methods such as CLARITY are applied [9]. These clearing methods can be combined with a wide variety of optical microscope technologies (e.g., two-photon excitation microscope, adaptive selective plane illumination microscope, confocal laser scanning microscope, and laser sheet fluorescent microscope) [10] to yield immunohistochemical images of the inner structures of the spheroids [11,12]. However, these clearing methods result in the loss of cellular or acellular morphological information that is important for cancer therapeutic strategies [7,13]. This is due to the loss of cellular or acellular substances such as cell membrane and lipid components that occurs as a result of these clearing methods. To obtain the cellular or acellular morphological information of the inner structures of the spheroids at the cellular level, sectioning is preferable [14,15,16,17,18]. For sectioning, clouded paraffin is typically used as an embedding resin for biological samples [19,20,21]. In addition to the morphological and immunohistochemical information, cross-sections of the spheroids provide mass spectra data regarding peptides and proteins that are often related to tumor behavior or etiologies underlying idiopathic diseases [22].

Conventional microfluidic devices are not suitable for sectioning. The typical material for microfluidic devices is polydimethylsiloxane (PDMS). PDMS is suitable for microfluidic devices due to its high processing accuracy, transparency, and gas permeability. However, the softness and elasticity of PDMS make the sectioning procedure difficult. During sectioning, PDMS is distorted due to its soft and elastic characteristics, and this distortion results in low accuracy of slice thickness and poor cross-sectional surface characteristics. Therefore, sectioning cannot be directly applied to the microfluidic devices created from PDMS. For in vitro experiments utilizing microfluidic devices, spheroids possessing blood vessels [6] or perfusing capillaries [23] were developed for use as more advanced in vitro experimental spheroid systems. Although these systems mimic the environment of real tumors, spheroids must be removed from microchannels using a biopsy punch [6]. This step destroys surrounding microstructures such as blood vessels. Therefore, there are potential requirements that must be met for the effective sectioning of spheroids used in microfluidic devices.

Here, we propose a microfluidic device created from embedding resin that enables us to section both the microfluidic device and the spheroids inside the device after the completion of in vitro experiments. The entire procedure, including the introduction, culture, embedding, and sectioning of spheroids in the microchannel, has been demonstrated. The sectioned slices were analyzed by acquiring bright-field and fluorescent images. To address both sectioning and transparency, we chose a kind of epoxy-based embedding resin with high transparency (EPOX), as the material used for the microfluidic devices. Through the use of an EPOX microfluidic device, spheroids were effectively cultured, embedded, and sectioned inside a microchannel to allow for the visualization of the inner structures and protein distributions of the spheroids.

## 2. Materials and Methods

### 2.1. EPOX Microfluidic Device

#### Design of the EPOX Microfluidic Device

A schematic illustration of the EPOX microfluidic device is shown in Figure 1a. The EPOX microfluidic device consisted of two layers that included an EPOX layer that possessed a channel with three trenches, and a PDMS layer has two ports connecting to the channel in the EPOX layer. The dimensions of the channel and trench were 1.4 mm × 600 μm (width, depth) and 1.4 mm × 400 μm (width, depth), respectively. These trenches functioned as traps for spheroids. The PDMS layer did not possess any microchannels and had only two ports (4 mm × 4 mm× 3 mm, length × width × height) that functioned as traps for bubbles in the medium. One port with one hole of 3 mm in diameter was used for the introduction of spheroids and the injection of EPOX to embed spheroids inside the device. The other port with two holes of 1 mm in diameter was used for the introduction of a variety of reagents (described in Section 2.3.2). The PDMS and EPOX layers should be separated to allow for the sectioning of spheroids after experiments using the microfluidic devices. A temporal seal is preferable to permanent bonding by the surface treatments such as oxygen plasma and vacuum ultra-violet light. For the temporal seal, a mechanical method using a clamp and screw was used.

The fabrication process is shown in Figure 1b. Briefly, PDMS (Silpot 184 W/C, Dow Corning Toray, Tokyo, Japan. Base polymer to curing agent ratio was 10:1 by weight) and EPOX (Technovit^®^ EPOX, Kulzer, Hanau, Germany. Base EPOX to curing agent ratio was 2:1 by weight) were casted on molds for the PDMS and EPOX layers and cured for 3 h at approximately 100 °C and 40 °C, respectively, in an air-vented oven. The cured PDMS and EPOX layers were then peeled from the molds. For the peeling of the EPOX layer, it was peeled off at approximately 100 °C. Holes for the inlet and outlet were punched on the PDMS layer. To enhance the adherence of spheroids onto the channel surface of the EPOX layer, 10% type-IC collagen (Wako, Osaka, Japan) was coated at 37 °C for 30 min. All components were assembled with holders using screws (Appendix A).

The polyacetal plate was milled to fabricate the molds and holders. The molds were polished using an abrasive compound (PiKAL Nihon Maryo-Kogyo, Tokyo, Japan) to remove the residual trace of milling. They were designed by three-dimensional computer-aided design software.

### 2.2. Spheroids and Reagents

#### 2.2.1. Formation of Spheroids

The human gastric carcinoma cell line N87 was purchased from the American Type Culture Collection (Manassas, VA, USA) to prepare the tumor spheroid. Suspensions of N87 cells were seeded in U-shaped bottom wells of a 96-well plate (Costar, Washington, DC, USA) to gather the cells at the bottom of the wells. Each well was filled with 150 µL of culture medium. Culture medium was prepared with 500 mL of Dulbecco’s Modified Eagle Medium (DMEM (High Glucose with L-Glutamine, phenol-red, and sodium pyruvate), Wako), 10% fetal bovine serum (Wako), and 100 UI/mL penicillin-streptomycin (Wako). The cell seeding number was 1.5 × 10^4^ cells/well. After cell seeding, the 96-well plate was stored in an incubator at 37 °C and 5% CO_2_ concentration for 9 days, with changing of the culture medium once every 3 days.

#### 2.2.2. Reagents

To evaluate cell viability, spheroids cultured both in the microfluidic device and in a 96-well plate were fluorescently stained using calcein-AM (Invitrogen) and propidium iodide (PI, Wako) at 37 °C and 5% CO_2_ for 2 h. Calcein-AM and PI were dissolved in phosphate-buffered solution (PBS, Wako) to adjust their concentrations to 0.5 and 1 mg/mL, respectively.

To embed spheroids, 4% paraformaldehyde (Nacalai Tesque, Inc., Kyoto, Japan) and 99.5% ethanol (Wako) were used for chemical fixation and dehydration, respectively.

To evaluate protein distribution within the cross-sections of spheroids, the spheroid slices were stained using fluorescent dye-labeled antibodies. In this report, HER2 and integrin were visualized. HER2 is a protein that expresses strongly on the N87 cell surface [23]. Anti-HER2 antibody conjugated to AF488 (Alexa Fluor^®^ 488 anti-human CD340 (erbB2/HER2) Antibodies, BioLegend, San Diego, CA, USA) and anti-integrin antibodies conjugated to AF555 (Anti-integrin α_v_β_5_ Antibody, Alexa Fluor^®^ 555 Conjugated, Bioss, Woburn, MA, USA) were used. The concentrations of anti-HER2 antibody and anti-integrin antibody were 5.0 and 3.0 µg/mL, respectively. Slices of a sectioned spheroid were soaked in a mixed solution of the antibodies at 37 °C for 2 h. Dyed slices were placed between the slide and cover glasses.

### 2.3. Experimental Procedure

#### 2.3.1. Image Acquisition and Quantification

Bright-field images were captured using a digital camera (EOS 6D, Canon, Tokyo, Japan) mounted onto an inverted microscope (IX73, Olympus, Tokyo, Japan). Confocal fluorescence images were captured using an electron multiplying charge coupled device camera (EMCCD, iXon Ultra, Andor, Tokyo, Japan) mounted onto an inverted microscope (IX83, Olympus). For the excitation light source used for the confocal microscopy, solid-state lasers (wavelength: 488 and 561 nm, TAC, Saitama, Japan) were chosen. The confocal fluorescence images were saved in 16-bit TiFF format, and fluorescence intensity was quantified using a capture software (iQ3, Andor).

To evaluate the EPOX biocompatibility, cell viability was examined. Green (indicating live cells) and red (indicating dead cells) fluorescence pixels were counted. The percentage of live cells compared to all cells of each spheroid was calculated using the following equation:(1)$$\mathrm{Viability}\;(\%)=\frac{\mathrm{Pixels}\;\mathrm{of}\;\mathrm{green}\;\mathrm{fluorescence}}{\mathrm{pixels}\;\mathrm{of}\;\mathrm{green}\;\mathrm{fluorescence}\;+\;\mathrm{pixels}\;\mathrm{of}\;\mathrm{red}\;\mathrm{fluorescence}}\times 100\;$$

To evaluate the immunohistochemistry, HER2 and integrin were stained. The distributions of green fluorescence (indicating HER2) and red fluorescence (indicating integrin) intensities were measured for each pixel. Both fluorescent intensities were normalized to the maximum intensities for comparisons among spheroids.

#### 2.3.2. Operation of EPOX Device from Embedding to Sectioning of Spheroids

The entire procedure describing the use of the EPOX device is shown in Figure 2. DMEM was introduced into the microchannel using a syringe pump (KDS200, kd Scientific, Holliston, MA, USA) (Figure 2a). Spheroids cultured in 96-well plates were transferred to the microchannel through a 3-mm-diameter hole (Figure 2b). The spheroids were moved toward the microchannel using gravitational force through manual tilting of the microfluidic device and dropped into the trenches. The microfluidic device was horizontally placed into an incubator to maintain it at 37 °C and 5% CO_2_ overnight. During this incubation, the spheroids adhered onto the surface of the microchannel. DMEM was flushed out of the system using PBS at the flow rate of 50 µL/min for 10 min using the syringe pump. To fix the spheroids inside the microchannel, 4% paraformaldehyde was introduced at a flow rate of 50 µL/min for 10 min and this was followed by storage at room temperature for 1 h (Figure 2c). To dehydrate the fixed spheroids, ethanol and deionized (DI) water were introduced. By controlling the ratio of the ethanol and DI water, a graded series of ethanol was prepared in the channel (e.g., 50%, 70%, 90%, 99.5% ethanol). For example, 50% ethanol was prepared when the flow rates of both ethanol and DI water were 50 µL/min. The flow rate of ethanol and the duration of the dehydration were 50 µL/min and 5 min at each concentration, respectively (Figure 2d). Ethanol is usually used for dehydration but too strong without any dilution. Biological samples should be gradually dehydrated by graded series of ethanol to prevent the sample from collapsing. The residual ethanol was removed by the introduction of air using a syringe pump, and the ethanol was evaporated in the air-vent oven at 37 °C for 3 min prior to embedding the spheroids in EPOX (Figure 2e). Without this complete evaporation, EPOX cannot be fully cured in the following steps. EPOX was manually introduced through the large hole using a syringe (Figure 2f). After the addition of EPOX into the microchannel, the microfluidic device was placed in the air-vent oven and cured at 37 °C for 4 h (Figure 2g). The EPOX layer was separated from the PDMS layer (Figure 2h) and was cut into a square 5 × 5 mm chip (Figure 2i). The chip was glued onto a specimen holder of a cryostat (HM550, Thermo Fischer) using EPOX. The glued chip was sectioned to a 10–20 µm thickness at room temperature (Figure 2j). Slices of the spheroid were collected on the slide glass (Figure 2k).

## 3. Results and Discussion

### 3.1. EPOX Biocompatibility

The biocompatibility of EPOX is important for performing biological experiments using spheroids within EPOX microfluidic devices. To analyze the toxicity of EPOX with regard to spheroids, the viability of cells in the spheroids was measured.

The viability of cells in the spheroids cultured in the EPOX device for 1 day after introduction was 92.4 ± 1.3% (N = 3), where N is the sample number and errors are standard deviation. In contrast, the viability of the cells in the spheroids cultured in the 96-well plate for 1 day after introduction was 94.0 ± 2.3% (N = 3). There was no appreciable difference between the viability of cells in the spheroids in the EPOX device and that in the 96-well plate.

### 3.2. Whole Slice Imaging of Sectioned Spheroids

Figure 3 shows the slices of a sectioned spheroid embedded in the EPOX microfluidic device. The embedded spheroid was sectioned from slices 1 to 16 in the order indicated in Figure 3a. The shape of the cross-section of the spheroid is elliptical rather than circular (Figure 3b). This is due to the adherence of the spheroid onto the surface of the microchannel covered by collagen as a scaffold to promote the adhesion of spheroids. The width of the spheroid was approximately 300 µm. All the slices were visualized by a bright-field microscope (Figure 3c). A white dotted line shows the outline of the shape of the trench (slice 1 in Figure 3c). The thickness of the slices was 18 µm and the total thickness was 288 µm. The diameter of the spheroid before sectioning and the total thickness of the sectioned spheroid were comparable. This result suggests that the spheroid embedded in the microchannel was well sectioned without removal from the microchannel. From the analysis of all the slices, we could obtain precise three-dimensional information for the spheroid.

### 3.3. Immunohistochemistry of Spheroids

To immunohistochemically analyze the distribution of the HER2 and integrin proteins throughout the spheroid, the slices of 10 µm in thickness were dyed in the mixed solution of HER2 and integrin antibodies. Figure 4 shows the distribution of HER2 and integrin within the cross-section of the spheroid. The slice was obtained from the center of the spheroid (white dash line in Figure 4a). In Figure 4b, no specific structures were observed. Strong expression of HER2 and integrin was observed near the surface (Figure 4c,d). Furthermore, high HER2 and integrin expression was also observed in proximity to the core. The expression level of HER2 and integrin outside the core was low with the exception of the surface of the spheroid. These features correspond to the protein distributions in N87 spheroids [24,25], and they suggest that the inner information of the spheroid was obtained. We believe that these structures within the spheroid may correspond to proliferative, dormant, and necrotic core regions [26]. The structure in the spheroid over 300 µm in diameter was difficult to visualize using the conventional analysis method involving the combination of a PDMS microfluidic device and a confocal microscope.

Structures and functions retained in conventional microfluidic devices (e.g., micro-pillar [6], valve [27], concentration gradient generator [28] etc.,) can be constructed thanks to the high processability of EPOX. We expect that the EPOX microfluidic device will achieve advanced in vitro experiments to spheroids such as angiogenesis with spheroids and high-throughput drug screening. The spheroids tested in the EPOX microfluidic devices can be sectioned, and then, the internal structures, especially hypoxia region in cancer [29,30], will be morphologically and immunohistochemically analyzed, which is difficult in the conventional methods using microfluidic devices and considerably important for biological and medical studies.

## 4. Conclusions

We developed the EPOX microfluidic device to perform in vitro experiments using a spheroid in a microchannel and to morphologically and immunohistochemically analyze the sectioned spheroid inside the microchannel. We achieved the embedding and the sectioning of the spheroids inside the microchannel. Using this microfluidic device, we obtained and observed slices of the spheroids. The inner structure and protein distribution of thick biological samples (over 300 µm) such as spheroids, organoids, and even tissues can be analyzed. The inner structures often include key information that is critical for understanding many vital phenomena and diseases. Thus, we believe that this microfluidic device will become a powerful tool that can be used to study thick biological samples.

## Figures and Tables

**Figure 1 micromachines-11-00480-f001:**
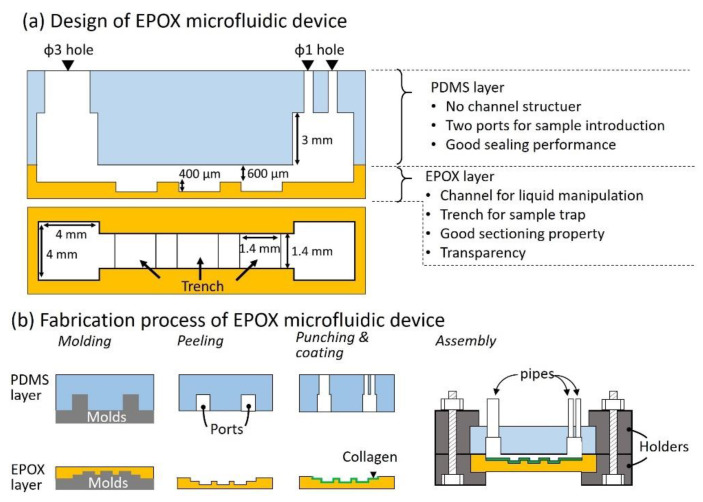
Design and Fabrication of the epoxy-based embedding resin with high transparency (EPOX) microfluidic device. (**a**) Schematic illustration and dimensions of the EPOX microfluidic device. A microchannel with trenches of the EPOX layer are used for traps of the spheroid. Two ports of the polydimethylsiloxane (PDMS) layer are used for the introducing of liquid, embedding resin and spheroids. (**b**) Fabrication process of the EPOX microfluidic device. Molding of PDMS and EPOX layers.

**Figure 2 micromachines-11-00480-f002:**
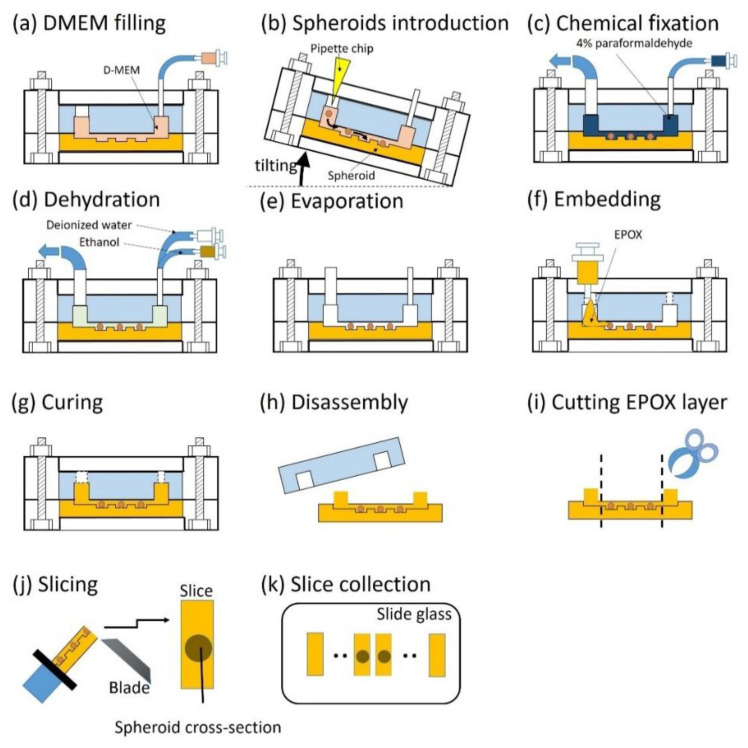
The operation of the EPOX device from embedding to sectioning of the spheroids. (**a**) DMEM filling. (**b**) Introduction of the spheroids using gravitational force by tilting. (**c**) Chemical fixation of the spheroids. (**d**) Dehydration of the spheroids using a graded series of ethanol. (**e**) Evaporation of the residual ethanol. (**f**) Injection of EPOX. (**g**) Curing of EPOX. (**h**) Device disassembly. (**i**) Cutting of the EPOX layer. (**j**) Slicing of the EPOX device by cryostat. (**k**) Slice collection.

**Figure 3 micromachines-11-00480-f003:**
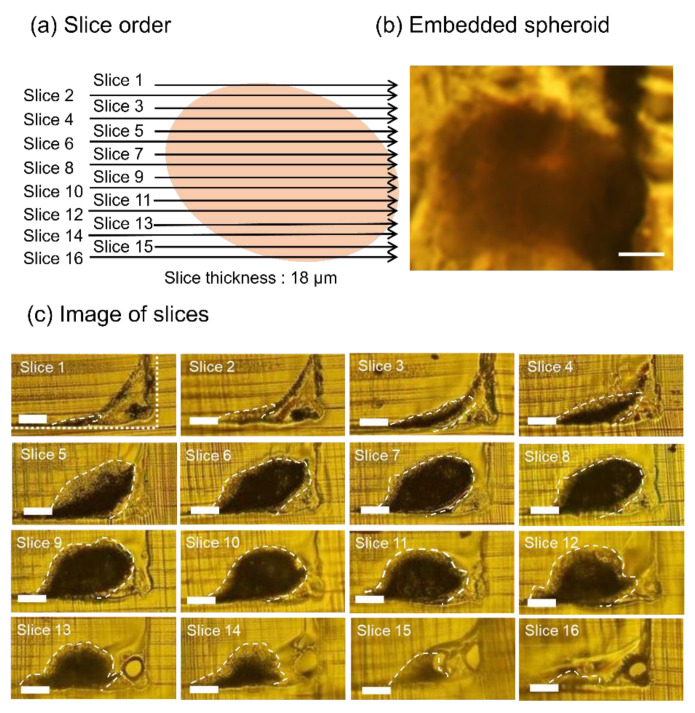
Images of slices of the sectioned spheroid. (**a**) Schematic illustration of the embedded spheroid. Slice orders and positions are described. (**b**) Image of the embedded spheroid. (**c**) All the slices of the sectioned spheroid embedded in the EPOX microchannel. The dark area corresponds to the spheroid in slices surrounded by white dash lines. Scale bar; 100 µm.

**Figure 4 micromachines-11-00480-f004:**
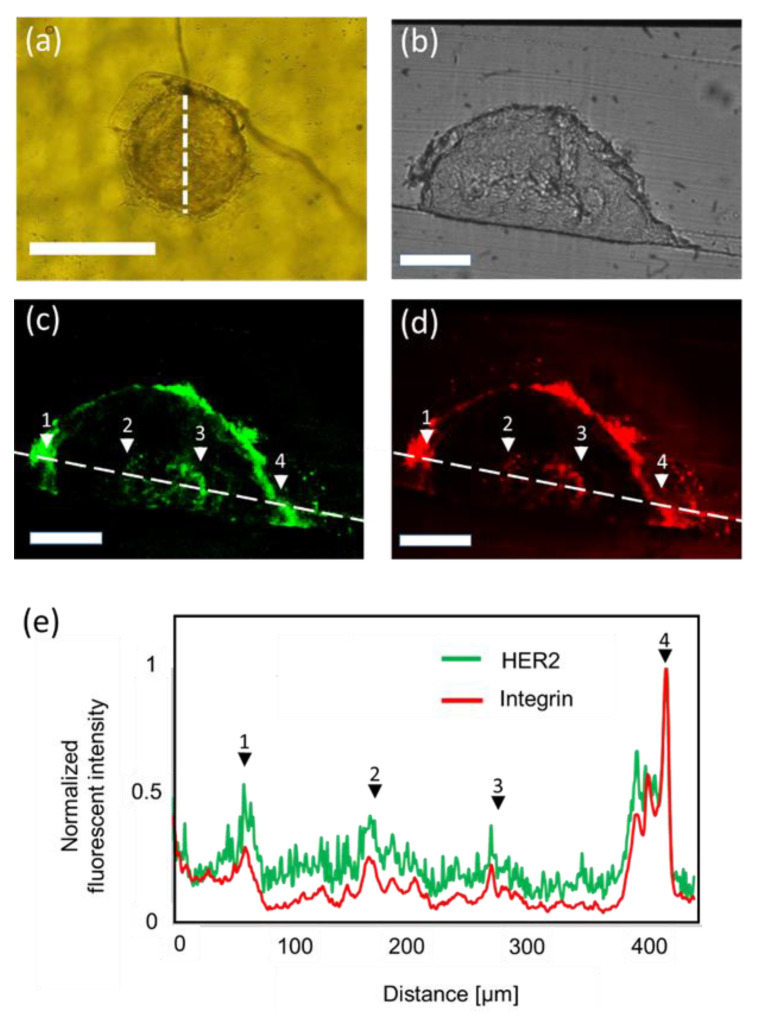
Immunohistochemical analysis of a slice of the sectioned spheroid. (**a**) Image of an embedded spheroid. White dash line indicates the location of the obtained slice. Scale bar: 300 µm. (**b**) Bright-field image of the slice of the sectioned spheroid. (**c**) Distribution of HER2 inside the spheroid. (**c**) Distribution of integrin inside the spheroid. Scale bar: 100 µm. (**e**) Normalized fluorescent intensity against the position along the white line of (**c**,**d**). Black arrows indicate the peak of fluorescent intensity and correspond to the white arrows of (**c**,**d**).

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
