# Peer review of "Slicing Spheroids in Microfluidic Devices for Morphological and Immunohistochemical Analysis"

_micromachines, 2020, doi:10.3390/mi11050480_

Round 1

Reviewer 1 Report

This paper by Kuriu et al. reported an embedding resin microfluidic device for sectioning of spheroid inside. This manuscript is technically sound and generally well-written. It should be of interest to readers of Micromachines. The reported method is useful for dealing with thick biological samples. This reviewer recommends publication in Micromachines after the following minor comments are addressed in the revised manuscript.

  1. Please check for grammar errors all through the manuscript. For example, Line 23: while areas intermediate layer between the surface and core possessed…
  2. Is it possible to show the “tilting of the microfluidic device” in Figure 4(b)?
  3. Line 171: CO2.
  4. In section 3.1, what does “N = 3” mean? What is N? Where do the errors come from (averaged over what? standard deviation or standard error of the mean?)?
  5. In Figure 5(c), is it possible to indicate all sectioned spheroids in cross-sections by white dash lines in all panels (not just 1, 2, 15, and 16)?
  6. For dehydration, what are the purposes of controlling the ratio of the ethanol and DI water (e.g. 50% ~ 99.5% ethanol)?
  7. Are the results obtained in this paper (with the microfluidic device) comparable to those obtained in other conventional methods, especially the immunohistochemistry part?

Author Response

We appreciate Reviewer #1 for taking the time to review our manuscript and give us his/her valuable comments. We have considered all the comments and have made appropriate changes to the manuscript.

Reviewer 2 Report

1) Title should be simplified, the term "embedding resin" is too specific and should be removed. Maybe: "Slicing the microfluidic device enables spheroid analysis"

2) Similarly, the Abstract should be simplified a bit (last 3-4 sentences). It is currently hard to understand how "resin", "spheroids" and microfluidics are combined.

3) Fig 1 is lacking i) explanations, detail, legend on the figure, ii) proper caption explaining what different parts of the chip do and iii) what parts are EPOX vs PDMS

4) Suggest merging together Fig1 and 2, it is easier to understand then

5) Fig 3 does not fdeel that important: i) move it to Supplementary or ii) merge it with Fig1+2

6) What is EPOX? Is it a bird, is it a plane, is it a chemical, is it a acronym?

7) The authors do not discuss the potential analytical application of their device. I presume that the main aim of microfluidics in spheroid analysis is NOT just to provide a "channel" for slicing, but to provide a tool for controlled introduction and removal of reagents, buffers, drugs etc. Authors should write a solid discussion paragraph about potential uses of their device: what sort of experiments are done and how does microfluidics help in spheroid analysis? Not just for slicing, but for the whole experiment.

Author Response

We appreciate Reviewer #2 for taking the time to review our manuscript and give us his/her valuable comments. We have considered all the comments and have made appropriate changes to the manuscript.
